# Fear of an unprecedented, invisible enemy: Difficulties experienced in establishing criteria for the release of COVID-19 patients from isolation in a Japanese University Hospital

**Hideharu Hagiya**⃝*, **Kou Hasegawa, Fumio Otsuka**

Department of General Medicine, Okayama University Graduate School of Medicine, Dentistry and Pharmaceutical Sciences, Okayama, Japan

* hagiya@okayama-u.ac.jp

**Data Availability Statement:** All relevant data are within the paper and its Supporting Information files.

## Abstract

### Introduction

The novel coronavirus disease 2019 (COVID-19) has emerged as a global pandemic, and the United States and European authorities established criteria for the release of COVID-19 patients from isolation in October 2020. However, a huge discrepancy exists between the hospital-discharge protocol for COVID-19 patients and the release of patients from in-hospital isolation. Our initially proposed criteria for in-hospital release from isolation was not adhered to by healthcare workers (HCWs) due to prevailing concerns regarding disease infectivity. Herein, we report difficulties encountered in attempting to establish a common understanding of the management of emerging infections.

### Methods

We performed a Google Form-based questionnaire survey targeting HCWs from Okayama University Hospital, Japan, via e-mail on January 21–28, 2021. The anonymous investigation required respondents to provide information regarding their background as well as perceptions regarding the requirement, level of understanding, and readiness for developing release criteria.

### Results

We obtained 150 eligible responses, including 57 (38.0%) from medical doctors and 53 (35.3%) from nurses. Most HCWs managing COVID-19 patients advocated for the implementation of the criteria, whereas those not working in that capacity did not ($p<0.001$). Over half of the HCWs indicated discomfort at seeing COVID-19 patients transitioning to general management even after meeting the criteria.

### Conclusions

It was challenging to establish a common understanding regarding the ideal criteria for in-hospital release of COVID-19 patients from isolation in our hospital. The dissemination of

**Funding:** The author(s) received no specific funding for this work.

**Competing interests:** The authors have declared that no competing interests exist.

our experiences and multifaceted discussions with HCWs would be of great value as a countermeasure against the emergent pandemic.

## Introduction

The novel coronavirus disease 2019 (COVID-19) wreaked worldwide havoc in 2020. The cumulative number of infected people worldwide surpassed 100 million in January 2021 [1], reflecting a greater medical, public, and social impact than the Spanish flu epidemic a century ago. In the first half of 2020, the disease was classified as a category 2 infectious disease according to the Act on the Prevention of Infectious Diseases and Medical Care for Patients with Infectious Diseases (the Infectious Diseases Control Law) in Japan. Currently, in addition to COVID-19, acute poliomyelitis, tuberculosis, diphtheria, severe acute respiratory syndrome, middle east respiratory syndrome, and avian influenza (H5N1, H7N9) are classified into this disease category. For a patient with these diseases, a prefectural governor can order hospitalization for the purpose of infection prevention and control.

A large disparity exists between the discharge protocol for COVID-19 patients and the release of patients from the isolation area to the general wards in a hospital. At that time, the Japanese government set the isolation period for 10 days after the disease onset, if 3 days have passed since the series of symptoms were alleviated. Regardless of whether they are directly engaged in treating patients with COVID-19 or not, it is evident to a greater or lesser extent that healthcare workers (HCWs) feel that the idea of accepting recently quarantined COVID-19 patients into general wards is discomforting, unacceptable, and objectionable, even if they meet the criteria for in-hospital release from isolation. These resentful sentiments are understandable, and we deliberately postponed the proposal of any criteria to the end of 2020 to avoid unnecessary conflicts in our hospital. However, a third wave surfaced throughout Japan at that time [2], and the number of hospital admissions, including severe cases, surged remarkably, thus compelling us to establish criteria for in-hospital release of COVID-19 patients from isolation.

In our hospital, every hospital standard for infection control measures against COVID-19 had to be reviewed by responsible persons of each department, and accepted by hospital executive office. At the beginning of January 2021, we drafted a first version of the criteria for in-hospital release from isolation based on those introduced by the United States and European Centers for Disease Control and Prevention (CDCs) in October 2020 [3, 4]; briefly, 10-days isolation after the disease onset. Despite persistent requests and appeals for general opinions regarding the releasing criteria, no consistent response was obtained from any the responsibles. Thus, the criteria proposed by members of the infection control team were adopted by the executives and launched at a general meeting conducted in mid-January 2021. However, these criteria were neither readily accepted nor adhered to by HCWs working in COVID-19 wards, leaving us with the mandate to further review and recompose the criteria. To fully understand the views and sentiments of our medical staff regarding the criteria to be adopted for in-hospital release from isolation, we elicited their opinions via an anonymous questionnaire. Herein, we report the results of the investigation, including the difficulties we experienced in establishing a common understanding of the emerging infection.

## Materials and methods

In this study, we conducted a Google Form-based questionnaire survey targeting HCWs at Okayama University Hospital in Japan. Through email, an anonymous questionnaire form

was disseminated to all hospital staff (approximately 2,000 people) on January 21–28, 2021. The questionnaire asked the respondents to provide background information regarding the following: (i) their occupation (physician, nurse, rehabilitation therapist, laboratory technician, clinical engineer, pharmacist, radiographer, or other), (ii) the number of years they have worked in their profession (1–2 years, 3–5 years, 6–10 years, 11–15 years, 15–20 years, or more than 20 years), and (iii) whether or not they worked in a ward treating COVID-19 patients. The following questions were posed to them for the purpose of investigating the requirement, level of understanding, and readiness for developing the criteria: [Question 1] "Are you in any trouble in the absence of criteria for in-hospital release from isolation at our hospital?"; [Question 2] "Do you agree with the development of criteria for in-hospital release from isolation?"; and [Question 3] "How would you feel if COVID-19 patients are placed under general management without any particular infection prevention measures after meeting the criteria for in-hospital release from isolation?". To prevent repeated answers from a same person, we set the Google Form survey not to accept two or more responses from the same identification.

Validity of these questions had not been piloted beforehand since they were not for the research purpose in the first place.

The study was approved by the Ethics Committees of Okayama University Graduate School of Medicine, Dentistry and Pharmaceutical Sciences and Okayama University Hospital (no. 2105–001) in accordance with Declaration of Helsinki. Informed consent from individual HCWs was not necessary due to the anonymization of data.

## Results

We obtained 150 eligible responses, of which 57 (38.0%) from medical doctors and 53 (35.3%) from nurses were included (**Table 1**). In terms of working years, 43 respondents (28.7%) had been working for $\geq$ 21 years and 50 (33.3%) for 11–20 years, suggesting that approximately 60% of the respondents occupied mid-level to management positions. The proportions of

**Table 1. Backgrounds of respondents (N = 150).**

| Occupation | Number (%) of respondents |
|---|---|
| Doctor | 57 (38.0%) |
| Nurse | 53 (35.3%) |
| Rehabilitation therapist | 13 (8.7%) |
| Laboratory technician | 7 (4.7%) |
| Clinical engineer | 7 (4.7%) |
| Pharmacist | 5 (3.3%) |
| Radiographer | 3 (2.0%) |
| Other | 5 (3.3%) |
| **Years of professional service** | |
| 21 years and more | 43 (28.7%) |
| 11 to 20 years | 50 (33.3%) |
| 6 to 10 years | 28 (18.7%) |
| 3 to 5 years | 14 (9.3%) |
| 1 to 2 years | 15 (10.0%) |
| **Engaged in treating COVID-19 patients in the isolation ward?** | |
| Yes | 34 (22.7%) |
| No | 77 (51.3%) |
| No response | 39 (26.0%) |

HCWs who were treating COVID-19 patients at the time of the survey and those who were not were 22.7% and 51.3%, respectively.

The results of responses regarding the need for criteria for in-hospital release from isolation are summarized in **Table 2**. About 1/4 of the respondents confirmed that they had trouble managing COVID-19 patients, and more than 80% of HCWs indicated that they required the criteria in our hospital. However, over half of the respondents indicated that they were apprehensive regarding the risk of infection upon seeing recently quarantined COVID-19 patients, who had met the criteria, in the general wards.

Of the 42 respondents who faced challenges due to the absence of the criteria, 20 (47.6%) were working in COVID-19 wards and 8 (19.0%) were not (**Table 3**). On the other hand, among the 108 respondents who had not been involved in any troubles in the absence of the criteria, 14 (13.0%) were working in COVID-19 wards and 69 (63.9%) were not. Fisher's exact test revealed a significant difference between the two variables (odds ratio [95% confidential interval]; 11.0 [4.1–38.3]: $p < 0.001$).

## Discussion

Our anonymized questionnaire survey demonstrated that most HCWs in our hospital advocated for the development of criteria for in-hospital release from isolation. However, about 3/4 of the HCWs practically faced no troubles due to the absence of criteria, and over half of them registered discomfort on seeing recently quarantined COVID-19 patients under general management even after meeting the criteria. These could be the main reasons underlying the failure to gain an understanding of the initial criteria launched in mid-January 2021 in our hospital. The tendency not to express individual opinions in public may be a peculiarity among Japanese people including HCWs [5], which makes it difficult for us to be aware of their real impressions and considerations. In particular, this could be the case regarding COVID-19, an emerging infectious disease, whose actual entity and infectivity are not yet fully uncovered.

Our first proposal regarding the criteria for in-hospital release from isolation, which was gleaned from the United States and European CDCs [3, 4], was based on evidence of the disappearance of replication-competent viruses within a certain period [6, 7]. However, this criterion was not adhered to by HCWs managing COVID-19 patients due to concerns regarding potential infectivity and safety. Despite the scientific evidence-based recommendations, there was a latent, difficult-to-verbalize fear of the unprecedented virus among the

**Table 2. Questions and answers regarding the in-hospital release of COVID-19 patients from isolation.**

| Question 1. Are you in any trouble in the absence of the criteria for the in-hospital release from isolation at our hospital? | |
| --- | --- |
| Yes, I am in trouble. | 42 (28.0%) |
| No, I am not in any trouble at all. | 108 (72.0%) |
| **Question 2. Do you agree to the development of the criteria for the in-hospital release from isolation?** | |
| Yes, it is needed. | 121 (80.7%) |
| No, it is not needed. | 1 (0.7%) |
| Not sure. | 28 (18.7%) |
| **Question 3. How would you feel if COVID-19 patients are placed under general management without any particular infection prevention measures after meeting the criteria for the in-hospital release from isolation?** | |
| We can manage the patients fearlessly | 39 (26.0%) |
| We still worry about the risk of infection | 87 (58.0%) |
| Others | 24 (16.0%) |

**Table 3. The relative need for isolation-release criteria among healthcare workers managing COVID-19 patients and those who are not.**

|  | Yes, I am in trouble (N = 42) | No, I am not in any trouble at all (N = 108) | *p* |
|---|---|---|---|
| Working in COVID-19 ward | 20 (47.6%) | 14 (13.0%) | <0.001 |
| Not working in COVID-19 ward | 8 (19.0%) | 69 (63.9%) |  |
| No response | 14 (33.3%) | 25 (23.1%) |  |

Fisher's test was performed.

Responses to Question 1, "Are you in any trouble in the absence of isolation releasing criteria at our hospital?" are classified according to their workplaces.

HCWs regarding the risk of infection. This was undoubtedly a solid barrier to their ready acceptance of the criteria. Another month ensued before the discussion matured, and we were eventually able to enforce the criteria on March 1, 2021. Before that, patients, especially those under critical care, were isolated for prolonged periods of time; in such cases, we continued with full precautions and complete isolation even up to over 60 days after disease onset.

There are many negative implications emanating from the absence of criteria for in-hospital release from isolation for the management of emerging infectious diseases. First, it imposes a practical burden on HCWs. The physical and emotional stress of providing medical care in a contaminated area (the so-called "red zone"), even while wearing personal protective equipment (PPE), is considerable. Second, a shortage of PPE is also a major concern. The longer the isolation period, the more the PPE needed. The wastage or overuse of PPE should be avoided to the greatest extent possible, especially in the midst of a prolonged epidemic. Third, the required consultation or rehabilitation are greatly limited in the absence of the criteria, resulting in adverse effects, such as a delay or failure in properly diagnosing other conditions at the right moment as well as prolonged recovery time before discharge and reintegration into society. In light of these observations, the enforcement of reasonable criteria with a high compliance rate is imperative, for which sufficient time for consideration, repeated discussion, and small group meetings would be required.

Limitations of the study should be mentioned. First, a sample size for the statistical evaluation was not performed before the questionnaire survey. Second, a selection bias was unavoidable because those who answered the survey would be comparatively interested in or involved with the criteria for in-hospital release from isolation. Also, a response rate of the questionnaire survey was not calculated in detail. Third, a generalizability of the results should be corroborated by future researches. Despite these points to be addressed, data shown here would be of value in understanding the difficulty in establishing criteria for the release of patients with the newly-emerged pathogens from isolation.

Emerging infectious diseases will surely appear in the future. In particular, in this age of globalization as a consequence of developed transportation systems, it is plausible to anticipate an even higher infection risk than before, as evidenced by history. The nature of the next pandemic to strike the world is unknown; however, as medical professionals, we need to make courageous, generally acceptable, field-oriented, and sustainable decisions based on data available at each moment, while having a balanced view of gradually emerging evidence and various opinions from multidisciplinary medical personnel. Our current knowledge base, established through our experience with the COVID-19 pandemic, is potentially beneficial in the event of another menace due to a newly emerging infectious disease.

## Supporting information

**S1 Data.**
(XLSX)

## Acknowledgments

We would like to thank Editage (www.editage.jp) for assistance in editing this manuscript.

## Author Contributions

**Conceptualization:** Hideharu Hagiya.

**Data curation:** Hideharu Hagiya.

**Formal analysis:** Hideharu Hagiya.

**Investigation:** Hideharu Hagiya.

**Methodology:** Hideharu Hagiya, Kou Hasegawa.

**Supervision:** Fumio Otsuka.

**Writing – original draft:** Hideharu Hagiya.

**Writing – review & editing:** Kou Hasegawa, Fumio Otsuka.

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
