## [Decision Letter · Decision Letter 0]

9 Feb 2022

PONE-D-22-01171Fear of an Unprecedented, Invisible Enemy: Difficulties Experienced in Establishing Criteria for the Release of COVID-19 Patients from Isolation in a Japanese University HospitalPLOS ONE

Dear Dr. Hagiya,

Thank you for submitting your manuscript to PLOS ONE. After careful consideration, we feel that it has merit but does not fully meet PLOS ONE’s publication criteria as it currently stands. Therefore, we invite you to submit a revised version of the manuscript that addresses the points raised during the review process.

ACADEMIC EDITOR: Please respond to the reviewer's comments.

We look forward to receiving your revised manuscript.

Kind regards,

Kensaku Kasuga

Academic Editor

PLOS ONE

Journal Requirements:

Additional Editor Comments:

Please respond to the reviewer's comments.

Reviewers' comments:

Reviewer's Responses to Questions

**Comments to the Author**

1. Is the manuscript technically sound, and do the data support the conclusions?

Reviewer #1: Partly

Reviewer #2: Yes

2. Has the statistical analysis been performed appropriately and rigorously? 

Reviewer #1: Yes

Reviewer #2: Yes

3. Have the authors made all data underlying the findings in their manuscript fully available?

Reviewer #1: Yes

Reviewer #2: Yes

4. Is the manuscript presented in an intelligible fashion and written in standard English?

Reviewer #1: Yes

Reviewer #2: Yes

5. Review Comments to the Author

Reviewer #1: Hideharu Hagiya et.al. performed a Google Form based questionnaire survey regarding the requirement, level of understanding, and readiness for developing a criterion of the release of COVID19 patients from in-hospital isolation to general wards. Their major finding was that more than half of health care workers not managing COVID19 patients are not in trouble in the absence of such a criterion while the others actually managing COVID19 are in trouble. To be honest, this obvious finding is not adding any scientific novelty, however this research is still worth publishing on the point highlighting the conflict in medical workers on the COVID19 management. The following point should be addressed for acceptance.

Introduction section

1. Please clarify what kinds of preventive measure are required for diseases of “a category 2 infectious disease according to the Act on the Prevention of Infectious Diseases and Medical Care for Patients with Infectious Diseases” in Japan

2. Briefly summarize “the discharge protocol for COVID19 patents” required by the Japanese government.

3. Also show “the criteria for in-hospital release from isolation” you made.

Result section

4. How many HCWs received request for this questionnaire survey? If the responder rate is relatively small (e.g. less than 50%), description about selection bias should be placed in the limitation paragraph.

5. I am also afraid the numbers of HCWs treating COVID19 patients (22.7%) seems a little high. The authors should clarify whether “HCWs treating COVID19 patients” means “HCWs currently treating COVID19” or “including HCWs who used to treat COVID19, but now works in general wards”.

6. It is possible that HCWs treating COVID19 patients tend to replay more actively than those in general wards. Comparing responder rates between the groups would be informative.

Discussion section

7. “The tendency not to express individual opinions in public may be a peculiarity among Japanese people”

It would be better to add a reference to support above description.

Reviewer #2: this study investigated about difficulties encountered in attempting to establish a common understanding of the management of emerging infections. they concluded The dissemination of their experiences and multifaceted discussions with HCWs would be of great value as a countermeasure against the emergent pandemic.

this is well conducted study.

6. PLOS authors have the option to publish the peer review history of their article (what does this mean?). If published, this will include your full peer review and any attached files.

Reviewer #1: No

Reviewer #2: No

---

## [Author Response · Author response to Decision Letter 0]

17 Mar 2022

10th/February/2022

Dr. Kensaku Kasuga

Academic Editor

PLOS ONE

Ref: PONE-D-22-01171-R1

Fear of an Unprecedented, Invisible Enemy: Difficulties Experienced in Establishing Criteria for the Release of COVID-19 Patients from Isolation in a Japanese University Hospital

We hereby resubmit our above-named manuscript for reconsideration for publication in PLOS ONE. We have carefully considered all of the enclosed comments and addressed them as thoroughly as possible. Point-by-point responses to the reviewers’ comments are given below. The corrected sentences are noted with track changes in the revised version. 

We hope you will now find our revised manuscript finally acceptable for publication in PLOS ONE.

Sincerely yours,

Hideharu Hagiya, M.D., Ph.D.

Department of General Medicine, Okayama University Graduate School of Medicine, Dentistry and Pharmaceutical Sciences, 2-5-1 Shikata-cho, Kita-ku, Okayama 700-8558, Japan 

Tel: +81-86-235-7342 Fax: +81-86-235-7345

E-mail: hagiya@okayama-u.ac.jp

 

Comment from Reviewer #1

Hideharu Hagiya et.al. performed a Google Form based questionnaire survey regarding the requirement, level of understanding, and readiness for developing a criterion of the release of COVID19 patients from in-hospital isolation to general wards. Their major finding was that more than half of health care workers not managing COVID19 patients are not in trouble in the absence of such a criterion while the others actually managing COVID19 are in trouble. To be honest, this obvious finding is not adding any scientific novelty, however this research is still worth publishing on the point highlighting the conflict in medical workers on the COVID19 management. The following point should be addressed for acceptance.

Response

We greatly appreciate your effort to review our study. We have provided point-by-point comments below.

Introduction section

1. Please clarify what kinds of preventive measure are required for diseases of “a category 2 infectious disease according to the Act on the Prevention of Infectious Diseases and Medical Care for Patients with Infectious Diseases” in Japan

Response

Thank you for your question. In Japan, for a patient with these diseases, a prefectural governor can order hospitalization for the purpose of infection prevention and control. I have added this point into the introduction. (Page 4)

2. Briefly summarize “the discharge protocol for COVID19 patents” required by the Japanese government.

Response

We have additionally explained the isolation rule for COVID-19 patients in Japan as follows; “At that time, the Japanese government set the isolation period for 10 days after the disease onset, if 3 days have passed since the series of symptoms were alleviated.” (Page 4)

3. Also show “the criteria for in-hospital release from isolation” you made.

Response

Thank you for your suggestion again. We have revised and given the new sentences that explain the “the criteria for in-hospital release from isolation” in Page 5.

Result section

4. How many HCWs received request for this questionnaire survey? If the responder rate is relatively small (e.g. less than 50%), description about selection bias should be placed in the limitation paragraph.

Response

It is difficult to count up the exact number of people receiving our proposal on the survey. The number was estimated around 2,000, which was added in the methods section. Then this point was added as one of limitation points. (Page 6 and 11)

5. I am also afraid the numbers of HCWs treating COVID19 patients (22.7%) seems a little high. The authors should clarify whether “HCWs treating COVID19 patients” means “HCWs currently treating COVID19” or “including HCWs who used to treat COVID19, but now works in general wards”.

Response

We appreciate your comment. We meant “HCWs who were treating COVID-19 patients at the time of the survey“. The corresponding part was revised as such. (Page 7)

6. It is possible that HCWs treating COVID19 patients tend to replay more actively than those in general wards. Comparing responder rates between the groups would be informative.

Response

It is now impossible to figure out the responding rates between the two category responders. However, at the time of survey, we set the Google survey not to accept repeated answers from the same identification. We consider this could prevent the reply you are worrying about. We addressed this point in the method section. (Page 6)

Discussion section

7. “The tendency not to express individual opinions in public may be a peculiarity among Japanese people”

It would be better to add a reference to support above description.

Response

We have given the recent article regarding this sentence. (Page 10)

Comment from Reviewer #2

This study investigated about difficulties encountered in attempting to establish a common understanding of the management of emerging infections. they concluded The dissemination of their experiences and multifaceted discussions with HCWs would be of great value as a countermeasure against the emergent pandemic.

this is well conducted study.

Response

We greatly appreciate your effort to review our study. Thank you for your comment.

---

## [Decision Letter · Decision Letter 1]

29 Mar 2022

Fear of an Unprecedented, Invisible Enemy: Difficulties Experienced in Establishing Criteria for the Release of COVID-19 Patients from Isolation in a Japanese University Hospital

PONE-D-22-01171R1

Dear Dr. Hagiya,

We’re pleased to inform you that your manuscript has been judged scientifically suitable for publication and will be formally accepted for publication once it meets all outstanding technical requirements.

Kind regards,

Kensaku Kasuga

Academic Editor

PLOS ONE

Additional Editor Comments (optional):

Reviewers' comments:

Reviewer's Responses to Questions

**Comments to the Author**

1. If the authors have adequately addressed your comments raised in a previous round of review and you feel that this manuscript is now acceptable for publication, you may indicate that here to bypass the “Comments to the Author” section, enter your conflict of interest statement in the “Confidential to Editor” section, and submit your "Accept" recommendation.

Reviewer #1: All comments have been addressed

Reviewer #2: All comments have been addressed

2. Is the manuscript technically sound, and do the data support the conclusions?

Reviewer #1: Yes

Reviewer #2: Yes

3. Has the statistical analysis been performed appropriately and rigorously? 

Reviewer #1: Yes

Reviewer #2: Yes

4. Have the authors made all data underlying the findings in their manuscript fully available?

Reviewer #1: Yes

Reviewer #2: Yes

5. Is the manuscript presented in an intelligible fashion and written in standard English?

Reviewer #1: Yes

Reviewer #2: Yes

6. Review Comments to the Author

Reviewer #1: Thank you for the author's revision. All issues I proposed have been addressed. The manuscript deserves to be published.

Reviewer #2: this is well conducted study. and the authors responded the all reviewer's comment and revised manuscript.

7. PLOS authors have the option to publish the peer review history of their article (what does this mean?). If published, this will include your full peer review and any attached files.

Reviewer #1: No

Reviewer #2: No

---

## [Editor Report · Acceptance letter]

4 Apr 2022

PONE-D-22-01171R1 

Fear of an Unprecedented, Invisible Enemy: Difficulties Experienced in Establishing Criteria for the Release of COVID-19 Patients from Isolation in a Japanese University Hospital 

Dear Dr. Hagiya:

I'm pleased to inform you that your manuscript has been deemed suitable for publication in PLOS ONE. Congratulations! Your manuscript is now with our production department. 

Kind regards, 

on behalf of

Dr. Kensaku Kasuga 

Academic Editor

PLOS ONE